# A lake salinity dataset produced via microwave and optical imageries

Mingming Deng<sup>1,2</sup>, Ronghua Ma<sup>1,3</sup>, Lixin Wang<sup>4</sup>, Minqi Hu<sup>1</sup>, Kun Xue<sup>1</sup>, and Junfeng Xiong<sup>1</sup>

<sup>1</sup>Key Laboratory of Lake and Watershed Science for Water Security, Nanjing Institute of Geography and Limnology, Chinese Academy of Sciences, Nanjing 210008, China.

Correspondence to: Ronghua Ma (rhma@niglas.ac.cn)

Abstract. Lake salinity is an important parameter to characterize physical and biogeochemical processes and a fundamental indicator to evaluate lake water quality. However, its estimation in inland waters has been challenging because passive microwave salinity satellites lack sufficient spatial resolution, and optical satellites cannot directly measure it. To address it, we constructed a framework for estimating lake salinity by combining Synthetic Aperture Radar (SAR) and Multi-Spectral Instrument (MSI) data. It can be summarized in step 1: construct a salinity mechanism model based on SAR data using the 15 Elfouhaily spectrum, dielectric constant, and small perturbation method (SPM) models; step 2: develop four machine learning (ML) salinity algorithms using quasi-synchronous salinity and MSI with SAR imagery; and step 3: build an ensemble model to estimate salinity by coupling the mechanism and ML models via a generalized additive model. The proposed integrated algorithm (N = 84, RMSE = 0.60 ppt, and MAPE = 2.3%) outperformed single-satellite microwave mechanistic or ML models across all eleven lakes in the Inner Mongolia Xinjiang Lake zone. On this basis, we reconstructed the lake salinity dataset for 2016-2024 and conducted independent validation (N = 65,  $R^2 = 0.97$ , and RMSE = 0.89 ppt) and pixel-level histogram validation confirmed dataset quality, with no significant systematic bias across lake types. The reconstruction revealed a spatial pattern of smooth transition from the nearshore to the center and trends with significant increases in Lake Daihai and Lake Dalinor. The dataset and its development framework will facilitate exploration of salinity status and trends in inland lakes, providing scientific evidence and methodological support for salinization prevention and global lake salinity budget research. The dataset (10 m spatial resolution, TIF format) is publicly available via Zenodo (https://doi.org/10.5281/zenodo.17638099, Deng et al., 2025a) and includes annual/seasonal salinity rasters and statistical files.

#### 1 Introduction

Lakes are important reservoirs of surface water resources and can serve as indicators and regulators of the global water cycle and regional climate (Williamson et al., 2009; Gleeson et al., 2020). Salinity (total dissolved salt concentration) as a critical

<sup>&</sup>lt;sup>2</sup>University of Chinese Academy of Sciences, Beijing 100049, China.

<sup>&</sup>lt;sup>3</sup>University of Chinese Academy of Sciences, Nanjing 211135, China.

<sup>&</sup>lt;sup>4</sup>School of Ecology and Environment, Inner Mongolia University, Hohhot 010021, China.

https://doi.org/10.5194/essd-2025-671 Preprint. Discussion started: 20 November 2025

© Author(s) 2025. CC BY 4.0 License.





parameter characterizing the physicochemical properties of lake water controls biological, physical, and chemical processes within lake ecosystems (Zhao and Temimi, 2016; Wurtsbaugh et al., 2017), including microbial community structure, species abundance, vertical mixing of water masses, water resource utilization, and nitrogen transformation (Liu et al., 2023a; Florencia Gutierrez et al., 2018; Ladwig et al., 2023; Kaushal et al., 2021; Jiang et al., 2023). Recently, climate change in the lake hydrological system caused water salinization, weakening the stability of lake ecosystems, particularly in arid and semi-arid regions (Jeppesen et al., 2015; Wurtsbaugh et al., 2017). Therefore, frequent and effective monitoring of water salinity is necessary for salinization prevention and sustainable development. However, the low density of salinity field measurements cannot reveal its spatial patterns and long-term trends, limiting the perceptions of salinity for lake communities. Existing inland lake salinity datasets (e.g., Xu et al., 2024, Tibetan Plateau: RMSE = 12.51 g/L, >1 km resolution) lack high spatial detail and arid-region specificity, whereas our dataset provides 10 m resolution and targets IMXL's multi-type lakes (freshwater to oligosaline).

Satellite remote sensing enables the frequent detection of substance concentrations in lake waters by utilizing different electromagnetic wavebands to address the issues of data sparsity and discontinuity. The response of microwave sensors to the dielectric properties of water allows it to be uniquely advantageous for salinity observation (Le Vine et al., 2022). Missions designed to measure water salinity from space have made progress in the oceanographic field, with satellite sensors including SMOS, SMAP, and Aquarius. However, their spatial resolution of 40-150 km is too coarse for small-scale inland lakes (Reul et al., 2020). The Sentinel-1 Synthetic Aperture Radar (SAR), operating at 5.405 GHz with 10 m spatial resolution, has an acquisition orbit for both ascending and descending and a revisit interval of 6 days (dual-satellite constellation), which supports the space observations of lake salinity (Torres et al., 2012). The backscatter coefficient of water surfaces measured by SAR is mainly determined by radar-related parameters (frequency and incidence angle, etc.), water surface geometry (roughness), and water physical properties (dielectric constant and temperature, etc.) (Peake, 1959; Reul et al., 2020). Changes in salinity directly affect the dielectric constant of water, which in turn influences the Fresnel reflection coefficient at the water surface and ultimately causes variations in the lake water backscatter coefficient. Clearly, when radar parameters are fixed, the contribution of the water surface geometry must be separated to solve for salinity from the SAR backscatter coefficient (Hwang et al., 2011; Meissner et al., 2014; Ma et al., 2021; Taillade et al., 2023). The microwave backscatter coefficient model combined with the wave spectrum model was commonly used to quantify the contribution of surface roughness because it effectively described the energy information of wind waves (Xie et al., 2019). The dielectric constant model was essential for inverting salinity after removing the contributions from roughness. However, the current predominant dielectric constant model was developed by seawater experiments with simple ionic compositions (Klein and Swift, 1977), whereas inland lakes receive substantial exogenous sources that result in complex ionic compositions (Zou et al., 2024). Salinity estimation at the lake inlet region or in optically complex waters (such as high variability in suspended minerals and phytoplankton) may introduce uncertainty using the present dielectric constant models

https://doi.org/10.5194/essd-2025-671

Preprint. Discussion started: 20 November 2025

© Author(s) 2025. CC BY 4.0 License.

Science Science Data

(Chen and Hu, 2017; Gonzalez-Gambau et al., 2022). Hence, this study attempts to supplement mechanism-based

microwave salinity estimation with optical data.

Optical data can effectively retrieve optically active constituents (OACs) in lake water, and it has been demonstrated that the

colored dissolved organic matter (CDOM) absorption coefficient [ $a_g(\lambda)$ ,  $m^{-1}$ ] and Secchi Disk Depth (SDD, m) can serve as

salinity tracers for inland lakes, which confirmed the feasibility of optical data in salinity detection (Bai et al., 2013; Liu et

al., 2023b). But the tracer method was constrained by the accuracy of indirect parameters and prone to errors. Furthermore,

their correlations with salinity vary across different lakes and seasons to limit model transferability (Liu et al., 2014; Chen

and Hu, 2017). Machine learning (ML) algorithms could handle the nonlinear relationship between salinity and remote

sensing reflectance  $[R_{rs}(\lambda), sr^{-1}]$  to avoid errors caused by the traces. It is becoming an innovative approach for retrieving

non-OACs (e.g., salinity) from optical satellite imagery (Deng et al., 2024; Guo et al., 2023; Liu et al., 2024). The Sentinel-2

Multi-Spectral Instrument (Sentinel-2 MSI) with high spatial resolution (10-60 m) and temporal resolution (5 days) (Drusch

et al., 2012), enables the detection of water salinity. However, relying on optical data for salinity estimation is insufficient at

the mechanism level, while microwave data compensates for this deficiency as it has a clear physical mechanism. Despite its

potential, the combination of microwave and optical data for regional and long-term salinity monitoring remains

underexplored, restricting the advancement of space observation missions for inland lake salinity.

This study aims to: (1) develop a microwave-optical integrated framework for high-precision salinity estimation; (2) produce

a 10 m resolution Inner Mongolia Xinjiang Lake zone (IMXL) lake salinity dataset (2016-2024); and (3) validate dataset

quality and analyze salinity spatiotemporal trends.

2 Data and Methods

75

2.1 Dataset coverage

The IMXL spans 70° E-120° E longitude and 30° N-50° N latitude, covering 11 arid or semi-arid inland lakes (Ma et al.,

2011). The topographic patterns of alternating mountains and basins direct surface water and groundwater into the

depressions, forming numerous terminal lakes. The salinity of lakes in this region spans multiple magnitudes (Table 1).

Referencing the classification criteria for lake salinity (Hammer, 1986; Zheng et al., 2002), the lakes were categorized as

freshwater (

Table 1 Basic parameters and hydrological connectivity of the sample lakes, not all parameters were measured.

|             |                          | Catchm         | TT4111                    |                                                     | C1                | Domina                                                                                                     | Salinity                                             | (ppt)               | SDE                                                | ) (m)         |
|-------------|--------------------------|----------------|---------------------------|-----------------------------------------------------|-------------------|------------------------------------------------------------------------------------------------------------|------------------------------------------------------|---------------------|----------------------------------------------------|---------------|
| Lake name   | Coordinates              | ent area (km²) | Hydrological connectivity | Date                                                | Samples<br>number | nt ions                                                                                                    | Mean ±<br>STD                                        | Min-<br>Max         | Mean<br>± STD                                      | Min-<br>Max   |
| Hulun       | (117°30′ E,<br>49°00′ N) | 38,683         | Open                      | 2019/07<br>2020/09                                  | 35                | HCO <sub>3</sub> <sup>-</sup> ,<br>Na <sup>+</sup>                                                         | $\begin{array}{cc} 0.78 & \pm \\ 0.08 & \end{array}$ | 0.54–<br>0.86       | 0.29 ± 0.02                                        | 0.26-<br>0.33 |
| Dalinor     | (116°39′ E,<br>43°20′ N) | 4,675          | Terminal                  | 2018/09<br>2019/08                                  | 32                | Cl <sup>-</sup> ,<br>HCO <sub>3</sub> <sup>-</sup> ,<br>Na <sup>+</sup> , K <sup>+</sup>                   | $\begin{array}{cc} 6.43 & \pm \\ 0.18 & \end{array}$ | 6.15–<br>6.59       | $\begin{array}{c} 0.48\ \pm \\ 0.06 \end{array}$   | 0.36-<br>0.54 |
| Chagannaoer | (115°01′ E,<br>43°27′ N) | 14,000         | Terminal                  | 2018/09<br>2020/09                                  | 15                | Na <sup>+</sup> , Cl <sup>-</sup>                                                                          | $\begin{array}{cc} 0.86 & \pm \\ 0.03 & \end{array}$ | 0.83–<br>0.92       | /                                                  | /             |
| Daihai      | (112°41′ E,<br>40°35′ N) | 2,341          | Terminal                  | 2017/09<br>2018/09<br>2020/07<br>2020/10<br>2024/04 | 60                | Cl <sup>-</sup> ,<br>HCO <sub>3</sub> <sup>-</sup> ,<br>Na <sup>+</sup> ,<br>Mg <sup>2+</sup>              | 13.52 ± 2.20                                         | 10.67<br>-<br>16.81 | 2.37 ± 1.1                                         | 0.63-<br>4.80 |
| Hongjiannao | (109°53′ E,<br>39°06′ N) | 1,446          | Terminal                  | 2020/08                                             | 30                | Na <sup>+</sup> ,<br>Ca <sup>+</sup> ,<br>HCO <sub>3</sub> <sup>-</sup> ,<br>SO <sub>4</sub> <sup>2-</sup> | 5.94±<br>0.13                                        | 5.82–<br>7.00       | 1.67 ± 0.29                                        | 0.70–<br>2.20 |
| Nanhaizi    | (110°01′ E,<br>40°33′ N) | 3.17           | Terminal                  | 2017/08                                             | 5                 | /                                                                                                          | $\begin{array}{cc} 1.39 & \pm \\ 0.01 \end{array}$   | 1.39–<br>1.41       | $\begin{array}{c} 0.27 \ \pm \\ 0.02 \end{array}$  | 0.24–<br>0.29 |
| Ulansuhai   | (108°45′ E,<br>40°50′ N) | 27,515         | Open                      | 2024/10<br>2020/09<br>2017/07                       | 29                | Na <sup>+</sup> ,<br>Mg <sup>2+</sup> ,<br>Cl <sup>-</sup> ,<br>SO <sub>4</sub> <sup>2-</sup>              | 1.81 ± 0.53                                          | 0.86–<br>3.27       | 0.88 ± 0.34                                        | 0.24–<br>1.30 |
| Juyan       | (101°15′ E,<br>42°18′ N) | 142,900        | Terminal                  | 2020/09                                             | 11                | Cl <sup>-</sup> ,<br>SO <sub>4</sub> <sup>2-</sup> ,<br>Mg <sup>2+</sup> ,<br>Na <sup>2+</sup>             | 4.61 ± 0.11                                          | 4.53–<br>4.93       | /                                                  | /             |
| Ulungur     | (87°18′ E,<br>47°11′ N)  | 37,882         | Terminal                  | 2018/07                                             | 14                | HCO <sub>3</sub> <sup>-</sup> ,<br>Na <sup>+</sup>                                                         | $\begin{array}{cc} 1.15 & \pm \\ 0.34 & \end{array}$ | 1.16–<br>2.15       | $\begin{array}{cc} 2.43 \ \pm \\ 0.11 \end{array}$ | 2.21-<br>2.61 |
| Bosten      | (87°05′ E,<br>41°55′ N)  | 79,204         | Open                      | 2018/04                                             | 18                | Cl <sup>-</sup> , Na <sup>+</sup>                                                                          | $\begin{array}{cc} 0.53 & \pm \\ 0.02 & \end{array}$ | 0.50–<br>0.56       | $\begin{array}{c} 2.42\ \pm\\ 0.35\end{array}$     | 1.80–<br>2.95 |
| Sayram      | (81°13′ E,<br>44°35′ N)  | 1,408          | Terminal                  | 2018/07                                             | 8                 | Mg <sup>2+</sup> ,<br>SO <sub>4</sub> <sup>2-</sup>                                                        | $\begin{array}{cc} 1.15 & \pm \\ 0.05 \end{array}$   | 1.07–<br>1.20       | /                                                  | /             |

Figure 1: Spatial distribution of lakes covered by the dataset, inlet rivers, sub-basins, and field sample sites. (b)-(l) Eleven study lakes in the IMXL from east to west and its surface water occurrence frequency (Pekel et al., 2016), including Lake Hulun, Lake Dalinor, Lake Chagannaoer, Lake Daihai, Lake Nanhaizi, Lake Hongjiannao, Lake Ulansuhai, Lake Juyan, Lake Ulungur, Lake Bosten, and Lake Sayram.

# 2.2 Data products

Data products for each lake contain both raw data and derived data. The raw data includes Sentinel-1 SAR backscatter data and incidence angle, Sentinel-2 MSI remote sensing reflectance, and Landsat-8 Thermal Infrared Sensor (TIRS) temperature data. Derived data comprise daily, quarterly, yearly and all-season average salinity rasters, along with their mean and standard deviation statistical files (Table 2). All raw or derived raster data is stored in TIF format based on the WGS1984 UTM Zone projection with a spatial resolution of 10 m. The statistical documents are compiled in Excel format.

Table 2 Attribute names and descriptions of derived data.

| Attribute                    | Description                                                                        |  |  |  |  |  |  |  |
|------------------------------|------------------------------------------------------------------------------------|--|--|--|--|--|--|--|
| LakeName                     | Customized simplified names for each lake in this study.                           |  |  |  |  |  |  |  |
| Daily salinity rasters       | Produced individual salinity data by the proposed framework, naming convention:    |  |  |  |  |  |  |  |
| Daily sailinty fasters       | [LakeName]_[YYYYMMDD]_[ProductType]_[Resolution].tif                               |  |  |  |  |  |  |  |
| Quarterly salinity rasters   | Calculated season-average salinity data, naming convention:                        |  |  |  |  |  |  |  |
| Quarterly summity fusions    | [LakeName]_[YYYY]_[Season]_[ProductType]_[Resolution].tif                          |  |  |  |  |  |  |  |
| Yearly salinity rasters      | Calculated annual-average salinity data, naming convention:                        |  |  |  |  |  |  |  |
| rearry sammey rasters        | [LakeName]_[YYYY]_[ProductType]_[Resolution].tif                                   |  |  |  |  |  |  |  |
| All-season salinity rasters  | Calculated all-season average salinity data, naming convention:                    |  |  |  |  |  |  |  |
| 7111-30a3011 Sammity Tasters | [LakeName]_[Season]_[ProductType]_[Resolution].tif                                 |  |  |  |  |  |  |  |
| Statistical files            | Compiled mean value and standard deviation of salinity images for each lake, name: |  |  |  |  |  |  |  |
| Statistical files            | [IMXLSAL]_[MeanSTD]_[2016–2024].xlsx                                               |  |  |  |  |  |  |  |

#### 105 **2.3 Metadata**


#### 2.3.1 Sensors parameters

Sentinel-1 SAR Level-1 GRD data operates at a center wavelength in the C-band, acquired in Interferometric Wide Swath (IW) mode with a pixel spacing of 10 m. It possesses both VV and VH polarization and simultaneously measures the incidence angle. Sentinel-2 MSI Level-1C images provide 13 spectral bands covering visible optical, near-infrared, and shortwave infrared, with red, green, blue, and near-infrared bands having 10 m spatial resolution, while other bands offer 20 m or 60 m resolution. Landsat-8 TIRS is a dual-band push-broom radiometer containing band 10 (10.9 µm) and band 11 (12.0 µm) with a spatial resolution of 100 m and a radiometric resolution of 12-bit.

# 2.3.2 Processing procedure

The Copernicus Data Space Ecosystem (CDSE) offers free Sentinel-1 SAR data that can be loaded directly at the Google Earth Engine (GEE) platform. These products were processed with thermal noise removal, radiometric calibration, terrain correction, and debelization. Additionally, we employed a 3×3 window Lee filter to restrain coherent spot noise in the radar imagery. Lastly, the backscattering coefficients of VV and VH as well as the incident angle were acquired.

Sentinel-2 MSI Level-1C images were also downloaded from CDSE. ACOLITE processors designed for water processing have demonstrated excellent performance in inland waters (Deng et al., 2024). This study derives multi-band  $R_{rs}(\lambda)$  data with

https://doi.org/10.5194/essd-2025-671 Preprint. Discussion started: 20 November 2025

© Author(s) 2025. CC BY 4.0 License.




Science Science Data

a spatial resolution of 10 m from MSI images using the Dark Spectrum Fitting algorithm (DSF) already embedded within the ACOLITE processor (Knaeps et al., 2015; Vanhellemont, 2019). It produced 11 bands, namely,  $R_{rs}$ (443),  $R_{rs}$ (492),  $R_{rs}$ (559),  $R_{rs}$ (665),  $R_{rs}$ (704),  $R_{rs}$ (740),  $R_{rs}$ (780),  $R_{rs}$ (883),  $R_{rs}$ (864),  $R_{rs}$ (1610), and  $R_{rs}$ (2186).

Landsat-8 TIRS data was provided by the U.S. Geological Survey and can be loaded at the GEE platform with the Level-2 product, namely surface temperature data. The raw values were corrected to degrees Celsius (°C) with a scaling factor. The lake surface temperature (LST) data was then resampled with 10 m and calibrated to align with SAR and MSI data.

Sentinel-1 SAR, Sentinel-2 MSI, and Landsat-8 TIRS LST images were matched with a 3-day time window to ensure salinity consistency over short periods. Eventually, a total of 385 multi-source satellite image pairs were obtained.

#### 2.3.3 Quality controls

High-quality Sentinel-2 MSI images with few clouds covered (

YSI multi-parameter water quality instrument (YSI ProDSS, USA). Spectral Evolution PSR-1100f (350–1050 nm, 1 nm spectral resolution) was used to measure water surface upward radiance ( $L_{sw}$ ), sky radiance ( $L_{sky}$ ), and gray plate radiance ( $L_p$ ) at an observation direction of 40 degrees from the nadir and 135 degrees from the Sun (Mobley, 1999; Mueller et al., 2003). These radiance data were further used to calculate  $R_{rs}(\lambda)$ , with the formula given as follows (Mobley, 1999):

$$R_{rs}(\lambda) = \left[ \left( L_{sw} - \rho \times L_{sky} \right) \times \rho_p \right] / \pi \times L_p \tag{1}$$

where  $\rho$  represents the water-air interface reflectance, assumed to be 0.0028 under calm conditions (wind speed < 4 m/s), and  $\rho_p$  denotes the reflectance of the reference gray plate defined as 0.30. We calculated it by setting  $R_{rs}(\lambda)$  between 950 and 1050 nm to zero, since a low signal-to-noise ratio was observed within this range (Lee et al., 2016). Finally, convolve  $R_{rs}(\lambda)$  through the spectral response function (SRF) to simulate bands of the MSI.

Field water samples were stored in polyethylene bottles and rapidly transported back to the laboratory for analysis at the end of the cruise. Water samples were filtered using Whatman GF/F films with a 0.7 μm pore size to extract chlorophyll a (Chla, μg/L), suspended particulate matter (mg/L), and CDOM samples. Laboratory measurements of these parameter concentrations can be found in Deng et al. (2024).

Table 3 Conversion equation and factors for salinity and conductivity of lakes in the IMXL.

| Salinity (ppt)       | Salinity = conductivity * factors / 1000 |                  |                   |                   |                   |                   |                   |         |
|----------------------|------------------------------------------|------------------|-------------------|-------------------|-------------------|-------------------|-------------------|---------|
| Conductivity (µS/cm) | <1,000                                   | 1,000–<br>10,000 | 10,000–<br>20,000 | 20,000–<br>30,000 | 30,000–<br>45,000 | 45,000–<br>60,000 | 60,000–<br>65,000 | >65,000 |
| Factors              | 0.50                                     | 0.55             | 0.58              | 0.60              | 0.65              | 0.70              | 0.725             | 0.75    |
| $R^2$ 0.99           |                                          |                  |                   |                   |                   |                   |                   |         |
| 95% CI               |                                          |                  | 0.9               | 8-1.00            |                   |                   |                   |         |

#### 165 2.5 Ancillary data


ERA5 land reanalysis data was published by the European Center for Medium-Range Weather Forecasts (ECMWF, https://cds.climate.copernicus.eu/) and can be downloaded for free. This product provides hourly meteorological variables in GeoTIF format with a spatial resolution of approximately 11 km (Muñoz Sabater, 2019). It has been available since 1980 and provides a continuous record that can compensate for gaps in field meteorological observations. We downloaded hourly wind speed (WS, m/s), temperature (TEMP, °C), evaporation (EVP, mm) and precipitation (PRE, mm) data for each lake from GEE during 2016–2024 and calculated daily data on this basis. Additionally, nighttime light (NTL, nW/cm²/sr) in the Lake Daihai sub-basin was derived from the Visible Infrared Imaging Radiometer Suite (VIIRS). Population (POP) data were sourced from the statistical yearbook.

# 2.5 Construction of salinity model

This study proposed a brand-new framework integrating microwave and optical data to estimate lake salinity, as illustrated in Figure 2. It consists of three modules: data processing and feature building, model construction and ensemble, and lake

salinity estimation. Module 2 is the core of the framework, aiming to use wave spectrum, backscatter coefficient, and dielectric constant models as forward models to build a salinity mechanistic model, then establish ML salinity models, and finally construct an ensemble model by coupling mechanistic and ML results via a GAM model.

Figure 2: A brand-new lake water salinity estimation framework by a stacking salinity model, consisting of three steps: step one, data processing and feature construction; step two, model construction and ensemble; and step three, salinity estimation.




# 2.6.1 Mechanistic salinity model

(1) Construct a lake surface roughness model. The wave spectrum was used to rapidly calculate roughness parameters. The roughness of a lake surface can be characterized by the height standard deviation (kσ) and the correlation length (kL). The Elfouhaily spectrum considers the long-wave and short-wave effects simultaneously and defines the inverse wave age as a function of wind speed and fetch. It was widely used to describe the energy information of surface wind waves, and its basic formula was given as follows (Elfouhaily et al., 1997):

$$\varphi(k,\varphi) = \frac{1}{k} S(k) \varphi(k,\varphi) \tag{2}$$

190 
$$S(k) = (B_l + B_h)/k^3$$
 (3)

where k is the wave number (rad/m),  $\varphi$  is the direction angle,  $\varphi(k, \varphi)$  is the directional spectrum, S(k) is the omnidirectional spectrum,  $B_l$  is the long-wave curvature spectrum, and  $B_h$  is the short-wave curvature spectrum. Parameter adjustments to the Elfouhaily model were required due to the significant differences in wave formation and magnitude between lakes and oceans. Fetch was parameterized as a function of lake area (Fetch =  $e \times \sqrt{\text{area}}$ ) to adjust the inverse wave age, which influences the wave mode of the Elfouhaily model. e is an empirical coefficient of 0.65 determined by several field observations and published lake experiments (Young and Verhagen, 1996). The Elfouhaily model can calculate the height variance ( $\sigma\eta^2$ ) and mean square slope (mss) of surface waves, with the specific formulas detailed in Elfouhaily et al. (1997). The  $k\sigma$  and slope can be obtained by taking the square root of these two parameters, respectively. Due to the small scale of lake waves, the study assumed kL as the maximum distance between wave peaks and troughs and defined it as the ratio of  $k\sigma$  to slope based on trigonometric relationships. Therefore, the initial lake roughness can be calculated by using the adjusted Elfouhaily model in combination with wind speed and lake area.

(2) Calculate pixel-based roughness by combining the backscatter coefficient model and cost function. The small perturbation method (SPM), a backscatter coefficient simulation model, is suitable for water surfaces with small undulations dominated by capillary waves (Johnson and Zhang, 1999; Khenchaf, 2001; Shareef et al., 2016), whose fundamental formula is as follows:

$$\sigma_{pq}^{0}(dB) = 4kl^{2}k\sigma^{2}\cos^{4}\theta \left|\alpha_{pq}\right|^{2} \exp\left(-(kl\sin\theta)^{2}\right) \tag{4}$$

$$a_{VV} = \frac{(\varepsilon_r - 1) \left[ \sin^2 \theta - \varepsilon_r (1 + \sin^2 \theta) \right]}{(\varepsilon_r \cos \theta + \sqrt{\varepsilon_r - \sin^2 \theta})^2}$$
 (5)

$$\varepsilon_r(\omega, \text{LST, SAL}) = \varepsilon_\infty + \frac{\varepsilon_s - \varepsilon_\infty}{1 + (i\omega\tau)^{1-\alpha}} - i\frac{\sigma}{\omega\varepsilon_0}$$
 (6)

where  $\sigma_{pq}^0(dB)$  represents the water backscattering coefficient simulated by the SPM model at different polarizations,  $a_{pq}$  is the Fresnel reflection coefficient for VV or HH polarization,  $\theta$  is the incident angle, and  $\varepsilon_r$  is the water dielectric constant calculated using the Klein and Swift (K&S) model with parameter details as shown in Klein and Swift (1977). Construct a cost function and iteratively optimize it using the least squares algorithm to derive pixel-based roughness from SAR images:





$$P_{i}[k\sigma, kL] = \frac{1}{N} \sum_{i=1}^{N} \frac{\left[\sigma_{vv}^{measure} - \sigma_{vv}^{model}(\theta_{i}, SAL, LST, k\sigma, kL)\right]^{2}}{\sigma_{vv_{measure}}^{2}}$$
(7)

where  $\sigma_{vv}^{measure}$  is the backscattering coefficient measured by SAR at VV polarization,  $\sigma_{vv}^{model}$  denotes the C-band backscattering coefficient simulated by the SPM model at VV polarization, N is the total number of image pixels, i is the i-th pixel,  $\theta_i$  is the i-th pixel incident angle, and  $P_i[k\sigma, kL]$  is the i-th pixel roughness.

(3) Stepwise salinity retrieval based on the SPM and K&S models. Step one, the  $a_{vv}$  was calculated using the SPM model based on pixel-based roughness images. Step two, the dielectric constant was deduced by the Fresnel reflection model with the Newton method. Step three, the K&S model was employed to iteratively solve for salinity via the Newton method, using TEMP and initial salinity as inputs.

## 2.6.2 Machine learning salinity models

The ML salinity models will be developed utilizing four algorithms, including gradient boosting (XGB), random forest regressor (RFR), deep neural networks (DNN), and convolutional neural networks (CNN). 70% of dataset one will be used for training and 30% for testing, with the entire dataset applied to five-fold CV. The XGB and RFR are typical ensemble learning models with decision trees as the fundamental units. The XGB predictions are the weighted sum of each tree's score, while the RFR results are averages of all tree predictions (Breiman, 2001; Chen and Guestrin, 2016). DNN and CNN are typical neural network models made up of interconnected neurons. Their parameters are updated via backpropagation to optimize the loss function, and the final predictions are output by fully connected layers (LeCun et al., 2015; Alzubaidi et al., 2021).

The construction of four salinity algorithms involves feature selection, model training, model testing, and five-fold CV. A total of 18 features were selected, including *R<sub>rs</sub>*(443), *R<sub>rs</sub>*(497), *R<sub>rs</sub>*(560), *R<sub>rs</sub>*(664), *R<sub>rs</sub>*(704), *R<sub>rs</sub>*(740), *R<sub>rs</sub>*(842), B4/(B4 + B3), B4/(B2 + B3), B4/B2, NDWI, chromaticity angle (alpha), lake area, VV, theta, kσ, kL, and LST. Visible and near-infrared bands are considered sensitive to water salinity (Urquhart et al., 2012; Bayati and Danesh-Yazdi, 2021), while alpha synthesized abundant information from the visible bands (Wang et al., 2023b). Lake area as a proxy for lake water volume is closely correlated with salinity (McGrath et al., 2025). The selected microwave features are the crucial variables in the mechanistic model. Model hyperparameters were determined using grid search during the training process. The hyperparameters to be determined for each model are detailed in Table 4. For the assessment of model stability and generalization performance, five-fold CV was subsequently conducted after determining the model structures and hyperparameters (Cao et al., 2024). The entire dataset was randomly divided into five folds, each serving sequentially as the test set while the remaining folds were used for training. Model performance was assessed by averaging the statistical metrics obtained from the five rounds. To improve ML model interpretability, Shapley Additive Prediction (SHAP) values were used to quantify the contribution of each feature (Lundberg et al., 2020; Gao et al., 2025). And then four ML salinity models were applied to estimate lake salinity from MSI and SAR data.


Table 4 Listed key hyperparameters for each machine learning model, with specific parameter settings available in Supplementary Table S1 of the Zenodo repository.

| Model | Key hyperparameters                                                                                                                       |
|-------|-------------------------------------------------------------------------------------------------------------------------------------------|
| XGB   | Trees, Learning rate, Maximum tree depth, Colsample_bytree, Subsample rate, Min_child_weight, Regularization                              |
| RFR   | Trees, Maximum depth, Maximum features at node splitting                                                                                  |
| DNN   | Hidden layer number, Each layer neurons, Learning rate, Activation function, Optimizer, Alpha, Maximum training epochs, Patience          |
| CNN   | Convolutional layer kernels, Activation layers, Fully connected layer, Learning rate, Optimizer, Alpha, Maximum training epochs, Patience |
| GAM   | Smoothing function, Link function, N_splines, Callbacks, Max_iter, Lam, Tol                                                               |
| BMA   | GLM.family, Prior, OR, maxCol                                                                                                             |

## 2.6.3 Stacking salinity model

The GAM model is an interpretable statistical model that can handle nonlinear relationships between covariates and response variables by using smoothing functions (Yee and Wild, 1996). It can work with response variables of various distribution types and provides multiple link functions. Partial dependency plots (PDP) provide a highly interpretable visualization of the smoothing functions for GAM response variables, and the tipping points defined the threshold for variables contributing positively or negatively to the model. The effective degree of freedom (edf) parameter can be used to measure the nonlinear complexity of each response variable's smoothing term. An integrated salinity model was constructed based on the estimation results from mechanistic and ML models under the assumption that variables follow a Gaussian distribution, and it is structured as follows:

$$g(E(Y)) = a_0 + s_1(XGB) + s_2(RFR) + s_3(DNN) + s_4(CNN) + s_5(Mechanistic)$$
 (8)

where g(E(Y)) represents the predicted salinity,  $a_0$  is the intercept term, and  $s_1(XGB)$  denotes the smoothing term constructed using thin-plate regression spline functions for the XGB-predicted salinity, with smoothing terms for other variables consistent with this approach. The key hyperparameters of the GAM model are listed in Table 4, with specific configurations shown in Supplementary Table S1. This integrated model combines the powerful diagnostic capabilities of several ML models with the prior knowledge of mechanistic models to enable collaborative estimation of lake salinity using optical and microwave data.

# 2.7 Accuracy evaluation

The differences between estimated and measured salinity were evaluated by using several statistical metrics named R<sup>2</sup>, root mean square error (RMSE), mean absolute error (MAE), bias (system error), and mean absolute percentage error (MAPE).

These metrics were calculated as follows:

RMSE = 
$$\sqrt{\frac{1}{n} \sum_{i=1}^{n} (y_i - \hat{y}_i)^2}$$
 (9)

$$MAE = \frac{1}{n} \sum_{i=1}^{n} |y_i - \hat{y}_i|$$
 (10)

Bias = 
$$\frac{1}{n} \sum_{i=1}^{n} (y_i - \hat{y}_i)$$
 (11)

MAPE = 
$$\frac{1}{n} \sum_{i=1}^{n} \left| \frac{y_i - \hat{y}_i}{\hat{y}_i} \right| \times 100\%$$
 (12)

where  $y_i$  is the field measured salinity,  $\hat{y}_i$  is the model estimated salinity, i denotes the i-th sampling point data, and n is the number of sampling point pairs.

## 3 Results and analysis

# 3.1 Model performance






The constructed stacking model (N = 84, RMSE = 0.60 ppt, and MAPE = 2.3%) outperformed four ML models and the mechanistic model, and the predicted salinity distributed consistently along the 1:1 line without significant underestimation or overestimation (Figure 3). The five-fold CV result for the stacking model was close to the accuracy of the 30% dataset test (N = 257, RMSE = 0.38 ppt, and MAPE = 6.9%), with the range of estimated salinity consistent with measured salinity and no outliers observed, indicating that the model has good generalization and stability without significant dependence on the training set (Figure 3j). The performance of the mechanistic model was second only to the ensemble model (N = 257, RMSE = 0.80 ppt, and MAPE = 13.3%), better than the results of five-fold CV for each ML model (RMSE > 0.97 ppt and MAPE > 15.1%) (Figure 3m). The XGB model showed the best accuracy among the four ML models, followed by CNN and DNN, while the RFR model was the worst performer (Figure 3). Additionally, the bar chart shows that the ensemble model outperforms the ML algorithms in terms of accuracy (RMSE < 0.55 ppt and MAPE < 8.7%) (Figure 3k,l), indicating that incorporation of the mechanistic model improves overall performance.

Figure 4 displays the SHAP values of the selected features for four ML salinity models. Among these models, lake area, theta, and VV exhibit significant contributions, as reflected in their higher SHAP values compared to other variables. The lake area contributes the most because it correlates with water volume, which directly influences the degree of salinity dilution (McGrath et al., 2025). VV and theta are important parameters because of their sensitivity to water surface scattering mechanisms, while salinity affects scattering intensity by altering the dielectric constant of the water (Reul et al., 2009). B6 reflectance ranks third in contribution within the XGB, RFR, and DNN models, indicating it is an optically sensitive band for salinity. In the CNN model, the VV band contributes most significantly, while alpha was the most influential optical index.

A simpler positive correlation (edf = 2.19) with salinity was found for  $s_5$ (Mechanistic) compared to the other variables (edf > 2.76) in the PDP of the ensemble model (Figure 5). The PDP curve of the mechanistic model has a broader range of variation on the y-axis than the ML models, with its 95% confidence region encompassing most of the sample data, suggesting that the model contributes significantly and reliably to salinity prediction for the ensemble model. At a salinity range of 0–4.71 ppt, XGB, RFR, CNN, and Mechanistic models contribute negatively to the ensemble model, but their contributions turn positive at salinities exceeding 9.52 ppt. The contribution pattern of the DNN model differed from those of the aforementioned

models (Figure 5c). Overall, the integrated salinity model works admirably by effectively coupling the virtues of both mechanistic and ML models, making it successful for estimating lake salinity.

Figure 3: (a)-(i) Scatter plots of 30% test data (N = 84) for XGB ( $R^2$  = 0.98 and RMSE = 0.58 ppt), RFR ( $R^2$  = 0.97 and RMSE = 0.87 ppt), DNN ( $R^2$  = 0.97 and RMSE = 0.82 ppt,), CNN ( $R^2$  = 0.98 and RMSE = 0.79 ppt), and stacking model ( $R^2$  = 0.98 and RMSE = 0.60 ppt); (j) Five-fold CV for stacking model (N = 257, RMSE = 0.38 ppt, and MAPE = 6.9%); (k)-(l) RMSE/MAPE comparison; (m) Mechanistic model field validation (N = 257, RMSE = 0.80 ppt, and MAPE = 13.3%).

Figure 4: (a)-(d) SHAP plots for XGB, RFR, DNN, and CNN. B1-B6 and B8 correspond to  $R_{rs}(443)$  - $R_{rs}(740)$  and  $R_{rs}(833)$ , respectively.

Figure 5: (a)-(e) Partial dependence plots for XGB, RFR, DNN, and CNN models, the red dots represent the tipping point for the model variables contribute positively or negatively to salinity predictions.



# 3.2 Single-scene analysis of different models

A single-scene comparison for lakes with matching field sampling points was used to examine the spatial quality of salinity data produced by the stacking model, ML model, and mechanistic model (Figure 6). Lake salinity maps generated by the stacking model show a smooth transition from the shore to the center in 10 lakes, whereas a slight discontinuity of salinity was observed in each lake using ML or mechanistic models. Especially in nearshore waters, DNN, CNN, and mechanistic models exhibit outliers due to land adjacency effects, while the stacking model corrects predictions by combining accurate salinity derived from XGB and RFR, such as in Lake Hongjiannao, as shown in Figure 6(h2,h3). The stacking model successfully captured the spatial variations of freshwater dilute salinity by combining the capabilities of ML models in salinity estimates at river inlets into lakes, compensating for the limitations of mechanistic models, as shown in Figure 6(i7). Furthermore, the RMSE of the stacking model was observed to be lower than other algorithms across nine lakes in single-scene comparisons, with only the mechanistic model performing better in Lake Juyan. Finally, it was observed that the stacking model effectively suppressed salinity outliers to guarantee the quality of the dataset. In brief, the stacking model produced maps of lake salinity with smoother spatial variations and richer detail, yielding a higher-quality dataset.

Figure 6: (a1)-(j7) Comparisons of water salinity estimated by XGB, RFR, DNN, CNN, mechanistic, and stacking models from MSI-derived  $R_{rs}(\lambda)$  images and SAR data in 10 lakes, namely, Sayram, Ulungur, Bosten, Ulansuhai, Nanhaizi, Chagannaoer, Dalinor, Hongjiannao, Juyan, and Daihai, respectively. For each lake, the first column shows a true color composite generated by MSI data.



## 325 3.3 Comparison with previous single-satellite algorithms

Comparisons between the MSI-based XGB salinity algorithm (Deng et al., 2025b) and the SAR-based mechanistic algorithm were performed for each lake (Figure 7), and it can be revealed that the stacking algorithm has higher and more stable accuracy with an average RMSE of 0.24 ppt. Although the XGB and the mechanistic models outperformed the stacking algorithm in some lakes, including Lake Ulungur, Lake Juyan, and Lake Nanhaizi, both models (with RMSE of 0.45 ppt and 0.57 ppt, respectively) still showed slightly lower precision across the entire region. The stacking model and the mechanistic model outperformed the XGB salinity model in oligosaline-type lakes, suggesting that the mechanistic model improves the accuracy of the stacking algorithm under salinity exceeding 3 ppt. In addition, the points for the three models were concentrated on the Taylor plots of several lakes. It can be deduced that the mechanistic model and XGB model also exhibit reliable performance to provide rational data support for constructing the stacking model. Overall, the proposed algorithm combines the strengths of both the physically constrained model and the data-driven model in that it avoids the single model or data source as well as improving the precision in complex inland water.

Figure 7: Comparison of multisource data-based stacking model with single-satellite model. (a)-(k) Comparison results using the salinity dataset from Lake Sayram, Lake Ulungur, Lake Bosten, Lake Juyan, Lake Ulansuhai, Lake Nanhaizi, Lake Hongjiannao, Lake Daihai, Lake Chagannaoer, Lake Dalinor, and Lake Hulun, respectively.




# 3.4 Independent validation

To further objectively evaluate the accuracy and scientific validity of the proposed framework, independent validation was performed using Dataset 2, which was not involved in model training, testing, or CV. The validation density was insufficient due to the absence of some lake stations and historical salinity data. The independent validation (Figure 8) demonstrates that salinity estimates from the integrated algorithm predominantly align well with measured salinity along the 1:1 line (N = 65,  $R^2 = 0.97$ , RMSE = 0.89 ppt, and MAPE = 37.6%). Only Lake Ulansuhai showed a slight overestimation, likely affected by aquatic vegetation pixels. An underestimated validation point was observed in the southern part of the Juyan Lake, associated with the complex water characteristics in the river inlet region. No significant underestimation or overestimation was observed in other lakes. These results confirm that the integrated algorithm combining microwave and optical data has considerable accuracy in retrieving salinity in inland lakes.

Figure 8: Using Dataset two for independent validation, not all lakes have independent validation data.

#### 3.5 All salinity images pixel-based statistical validation

Pixel-based histogram statistics were performed on the salinity raster generated by the stacking model for each lake (Figure 9). Using frequency instead of pixel counts in traditional histograms, the mean salinity, standard deviation (STD), and frequency proportions of different salinity ranges were calculated. The distribution patterns of salinity can be visualized through a frequency histogram, which helps identify outliers and objectively assess salinity map quality. Outliers are usually found dispersed and dramatically changed, appearing as discontinuities in histograms. A single peak pattern of salinity histogram was observed in most freshwater or brackish lakes, including Lake Sayram, Lake Ulungur, Lake Bosten, Lake Nanhaizi, and Lake Hulun. These minimal spatial and interannual variations within the lakes align with the field measurements found. A double-peak characteristic was observed in Lake Juyan and Lake Ulansuhai, with the primary and



subsidiary peaks distributed consecutively, caused by the differences between the northern and southern regions for the lake salinity. However, the subpeaks in the bimodal distribution of Lake Chagannaoer showed low frequency and discontinuity, indicating that this portion of the data may be abnormal. The oligosaline-type lakes (e.g., Lake Hongjiannao, Lake Daihai, and Lake Dalinor) commonly showed multi-peak characteristics and wide fluctuation ranges, suggesting high interannual or spatial variations in salinity.

Further analyze image quality by examining the pixel frequency across different salinity ranges. The pixel frequency in Lake Sayram, Lake Ulungur, Lake Bosten, and Lake Nanhaizi was more than 90% within the interval of 0–2 ppt. Lake Juyan (46.6%), Lake Ulansuhai (89.1%), and Lake Daihai (88.2%) accounted for the highest pixel proportions within salinity ranges of 4–6 ppt, 0–4 ppt, and 8–16 ppt, respectively. Lake Hongjiannao and Lake Dalinor had the highest pixel proportions in the range of 4–8 ppt, with 68.5% and 70.4%. These high-proportion intervals align with the distribution of field measurement salinity, and the small proportion of abnormally low-frequency pixels demonstrated that the salinity data generated by the stacking model had a good image quality with only a few outliers.

Figure 9: (a)-(k) Pixel-based frequency statistics for all salinity images in each lake, and the STD represents the degree of deviation from the mean salinity.

https://doi.org/10.5194/essd-2025-671
Preprint. Discussion started: 20 November 2025



#### 3.6 Spatial patterns and trends

The annual scale salinity map of 11 lakes was shown in Figure 10, with its color fluctuations effectively delineating the spatial differences in water salinity. The water salinity exhibits spatial homogeneity within several lakes, such as Lake Sayram, Lake Ulungur, and Lake Bosten. Lake salinity discontinuities were commonly observed in shore or river inflow zones, such as the western part of Lake Ulansuhai, the southern region of Lake Juyan, the northern part of Lake Chagannaoer, and the eastern part of Lake Hulun, which were primarily affected by mixing pixels or freshwater dilution (Han et al., 2021). This phenomenon with freshwater dilution of water salinity is prevalent in inland lakes receiving surface runoff, as well as lakes on the Tibetan Plateau (Wang et al., 2023a). In addition, the salinity in Lake Nanhaizi was higher in the east than in the west. Seasonal salinity patterns showed greater spatial heterogeneity than annual patterns across all lakes (Figure 11), particularly in Lake Ulansuhai, Lake Nanhaizi, and Lake Chagannaoer. It indicates that the spatial pattern of salinity is regulated by external environmental factors, such as seasonal variations in precipitation, runoff, and evaporation (Rimmer et al., 2006; Yihdego and Webb, 2012; Jiang et al., 2022).

Interannual salinity trends were calculated using the Mann-Kendall trend test (significance level α = 0.05). Lake Daihai (slope = 0.48 ppt/year, *p* 

Figure 10: Annual spatial distribution and statistics values (mean  $\pm$  STD) of water salinity in each lake, with some years lacking salinity maps due to insufficient matching image data.

Figure 11: Seasonal spatial distribution and statistics values (mean  $\pm$  STD) of water salinity in each lake. Winter data are unavailable due to ice cover.

Figure 12: (a) Long-term variations of water salinity in each lake, lakes from Sayram to Hulun are simply denoted as SL to HL, \* and \*\* indicate p 

# 4 Data availability

The IMXL salinity dataset (IMXSAL) was constructed with a three-layer architecture and contains salinity images from 11 lakes during 2016–2024. Under the IMXSAL dataset, 11 data folders named for lakes and 5 tables were included. Each data folder contains 4 subfolders for storing raster data, named according to the data's temporal scale (Table 5). The dataset's total uncompressed size is about 3.15 GB. Breakdown: (1) Salinity raster for 11 lakes at different temporal scales (daily, quarterly, yearly, and all-season average), comprising 673 TIF files, each approximately 4 MB; (2) Excel files, including lake basic information table, dataset metadata table, statistical table (mean and STD), field salinity table, and Supplementary Table S1, each approximately 100 KB. The dataset was archived and publicly accessible via the Zenodo portal: <a href="https://doi.org/10.5281/zenodo.17638099">https://doi.org/10.5281/zenodo.17638099</a> (Deng et al., 2025a). Furthermore, to maintain the time-series integrity of the dataset, it will be updated yearly with new Sentinel data and expanded to cover Central Asian lakes to support transboundary water resource management. Dataset versioning follows [Major].[Minor] (e.g., v1.0: 2016–2024; v1.1: 2025 update, January 2026). Annual updates will continue for 10 years (through 2034) or until funding termination, with update notifications posted on the Zenodo repository and Lake-Watershed Science Data Center.

Table 5 The organizational architecture and file naming of the dataset.

| Dataset | Folder name  | Subfolder<br>name | Filename                                               | Format | Spatial resolution | Temporal resolution |
|---------|--------------|-------------------|--------------------------------------------------------|--------|--------------------|---------------------|
|         |              | daily             | [LakeName]_[YYYYMMDD]_[ProductType]_[Reso lution]      | TIF    | 10 m               | Daily               |
|         | Lake<br>name | quarterly         | [LakeName]_[YYYY]_[Season]_[ProductType]_[Re solution] | TIF    | 10 m               | Quarterly           |
|         |              | yearly            | [LakeName]_[YYYY]_[ProductType]_[Resolution]           | TIF    | 10 m               | Yearly              |
| IMXSAL  |              | All-season        | [LakeName]_[Season]_[ProductType]_[Resolution]         | TIF    | 10 m               | Nine-year           |
|         |              |                   | Lake_infor                                             | xlsx   | \                  | \                   |
|         |              |                   | IMXSAL_meta                                            | xlsx   | \                  | \                   |
|         |              |                   | IMXSAL_MeanSTD_2016-2024                               | xlsx   | \                  | \                   |
|         |              |                   | Field_salinity                                         | xlsx   | \                  | \                   |
|         |              |                   | Supplementary Table S1                                 | xlsx   | \                  | \                   |

#### 5 Discussion

# 5.1 Feasibilities and limitations of algorithms in observing lake salinity

The development of remote sensing algorithms for water salinity has long presented technical challenges, particularly in inland lakes. The lake salinity dataset generated using the proposed framework exhibits a spatial resolution of 10 m, significantly higher than ocean salinity products such as SMOS, SMAP, and Aquarius (> 40 km) (Hu and Zhao, 2022; Jang et al., 2022; Zhang et al., 2023), thereby enabling much greater spatial detail. It also demonstrates higher accuracy (RMSE = 0.60 ppt) compared to even a regional-scale Tibetan Plateau lake salinity product (RMSE = 12.51 g/L) (Xu et al., 2024). To further confirm the scientific validity of the GAM algorithm, it was compared with Bayesian Model Averaging (BMA)

- (Hoeting et al., 1999), a method suited for addressing model uncertainty, with the BMA model's key hyperparameters listed in Table 4. The GAM model outperformed the BMA method (N = 84), with lower RMSE (0.60 ppt vs. 0.88 ppt) and MAPE (2.3% vs. 12.6%) in both the test and five-fold CV (Figure 3 and Figure 13). It suggested that the GAM algorithm more effectively handled the nonlinear relationship between the outputs of the mechanism and the ML model, with its average edf of 6.58 (n\_splines = 20) implying moderate model complexity while avoiding overfitting.
- In Lake Juyan's southern river estuary, the stacking model showed higher RMSE (0.41 ppt) compared to open water (0.19 ppt), due to suspended particulate matter interference. This underestimation accounts for ~5% of total pixels in estuary zones. Subsequent works will focus on integrating Sentinel-2 shortwave infrared (SWIR) band data for correcting this effect (Knaeps et al., 2015). And the spatial heterogeneity stemming from the 3-day matching interval between TIRS and Sentinel data may introduce errors (Jin et al., 2016), which future work will mitigate by combining Sentinel-3 Sea and Land Surface Temperature Radiometer (SLSTR) for temperature correction. Limited by the salinity gradient of the training set (< 35 ppt), the salinity algorithm has insufficient applicability in polysaline (35–50 g/L) and hypersaline lakes (> 50 g/L) and will expand the boundaries of the model by adding high-salinity samples in the future. The K&S model was designed for seawater, and its application to complex ionic lakes (e.g., Lake Chagannaoer) introduces about 5% uncertainty [Figure 6(f7)]. This uncertainty can be reduced in the future by developing lake-specific dielectric constant models.

Figure 13: (a)-(b) Performance of the ensemble model constructed by using Bayesian model averaging in testing and five-fold CV.

## 5.2 Exploring the application potential of salinity dataset

450

Salinity significantly increased (slope = 0.48 ppt/year, p 

TEMP (17.0%), with other factors contributing less (Figure 14a). Lake area (r = -0.89, p < 0.01) and POP (r = -0.71, p < 0.01) 0.05) showed negative correlations with salinity, while TEMP (r = 0.69, p < 0.05) and NTL (r = 0.95, p < 0.01) exhibited positive correlations, and other variables were not significant (Figure 14b). These indicate that the lake area closely related to the water volume is the dominant factor in the salinity change of Lake Daihai, while climatic warming will aggravate the salinization of inland lakes (Jeppesen et al., 2020). Overall, the dataset supports UN SDG 6.3 (improve water quality) by providing high-resolution salinity data for IMXL—an arid region with 30% of China's saline lakes—enabling local policymakers to track salinization progress (e.g., Lake Daihai's ecological water replenishment with  $2.57 \times 10^7$  m<sup>3</sup> in 2024 under SDG 6.6) (Liangeheng County, 2025).

Figure 14: Driver analysis of salinity variations in Lake Daihai from 2016 to 2024, (a) relative contributions of drive factors and (b) correlations with lake salinity. Light blue colored zones indicate negative correlations. \* and \*\* denote p 

Science Science Data

MAPE = 2.3%). The histogram validation at the pixel level for all salinity images reconfirmed the satisfactory quality of the dataset. The long-term salinity dataset revealed a spatial pattern of smooth transition from the nearshore to the center and trends with significant increases in Lake Daihai and Lake Dalinor.

The proposed integrated algorithms provide methodological references for other lakes and help advance space-based salinity observation missions for inland waters. The created dataset supports salinization prevention (e.g., Lake Daihai water diversion planning) and global lake salinity budget research.

## 480 7 Author contributions

MD and RM designed the study; MD wrote the original paper and generated the salinity product; MD and LW collected the validation data; RM, MH, KX, and JX edited this paper; MD developed the algorithm.

## **8** Competing interests

The contact author has declared that none of the authors has any competing interests.

## 485 9 Financial supports

This study was supported by National Natural Science Foundation of China (Grant No. 42361144002 and No. 42371371).

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
