# Peer review of "A lake salinity dataset produced via microwave and optical imageries"

_Earth System Science Data, 2025_

## Author Comment (AC1)

**Point-to-point actions in response to Anonymous Referee #1**

Manuscript #: essd-2025-671

We sincerely thank the Referee #1 for their thoughtful and constructive comments. We have revised the manuscript accordingly. The detailed point-by-point responses are provided below (blue), and the corresponding revisions in the manuscript are marked in red.

**Response to the Reviewer 1:**

The manuscript I reviewed is already a revised version. I like it very much. Not only is the research original, but the data products are unique and significant for various stakeholders. To my knowledge no one has published similar datasets in the literature. It's a pleasant read and I also learned from this reading. I have only a few editorial comments, which can be made during galley proof corrections.

**Reply:** We are very grateful that you took the time to review our manuscript, and sincerely thank your interest in our articles! Your valuable comments provided great assistance for the improvement of our articles. We addressed each comment as follows.

1) Fig. 1. Need to explain (a) in the caption. Also explain the meaning of "frequency of surface water occurrence" and the cross symbols in each panel.

**Reply:** Thank you for your professional suggestions. Firstly, we have added an explanation for Figure 1(a) in the figure caption. Secondly, we have explained that the frequency of surface water occurrence refers to the normalized count of pixels that were identified as water in Landsat imagery from 1984 to 2021. Finally, we have described the cross symbols in each panel, specifying that they represent field salinity sampling points.

**(Page 5, lines 93-97)** "Figure 1: (a) Spatial distribution of sampled lakes covered by the dataset, inlet rivers, and sub-basins. (b)-(l) Eleven study lakes in the IMXL from east to west and its surface water occurrence frequency (Pekel et al., 2016), including Lake Hulun, Lake Dalinor, Lake Chagannaoer, Lake Daihai, Lake Nanhaizi, Lake Hongjiannao, Lake Ulansuhai, Lake Juyan, Lake Ulungur, Lake Bosten, and Lake

Sayram. Cross symbols in each lake denote the field salinity sampling points. The frequency of surface water occurrence refers to the normalized count of pixels that were identified as water in Landsat imagery from 1984 to 2021."

2) Fig. 2 caption: Please add one sentence similar to this: "The details of individual steps and the meaning of the symbols are described below."

**Reply:** Thank you for your pointing this out. We have added it.
**(Page 9, lines 181-183)** "Figure 2: A brand-new lake water salinity estimation framework by a stacking salinity model, consisting of three steps: step one, data processing and feature construction; step two, model construction and ensemble; and step three, salinity estimation. Details of individual steps and the meaning of the symbols are described below."

3) Figs. 3 and 13. Change "Measure Salinity (ppt)" to "Measured Salinity (ppt)" in all x-axis labels.

**Reply:** Thank you for your professional suggestions. We have revised it.
**(Page 14, lines 298-301; Page 27, lines 498-499)**

[Figure]

**Figure 3**

**Figure 13**

4) Fig. 6. I like this figure as it makes visual inspection straightforward. The lower salinity around the land-lake boundaries may be due to runoff, but what caused the high salinity points (reddish colors) in c7?

**Reply:** We greatly appreciate your recognition, and thank you for your thoughtful comments. Through observation and analysis of MSI natural color composite images and salinity maps derived from various models, we found that the high salinity in the land-lake boundaries of Lake Bosten was caused by DNN and CNN overestimates of salinity influenced by mixed pixels. This high value was not fully corrected by the integrated model. Additionally, we supplemented the explanation for the high salinity pixels appearing in Figure 6(c7) in Section 3.2.

[Figure]

MSI natural color composite images

[Figure]

Salinity maps derived from different models

**(Page 16, lines 314-316)** "However, salinity values in Lake Bosten's nearshore pixels were not fully corrected because the DNN and CNN models overestimate salinity influenced by mixed pixels (Figure 6(c4,c5,c7))."

5) Fig. 11. Put some space between column 3 and column 4, and between column 6 and column 7.

**Reply:** Thank you for your pointing this out. We have revised it.

**(Page 25, lines 404-406)**

[Figure]

**Figure 11**

6) Eq. (8) – what's the unit of g? Need to list.

**Reply:** Thank you for your thoughtful comments. We have clearly specified that the units of integrated model estimates are aligned with the input data with units of ppt.

**(Page 12, lines 257-258)** "where g(E(Y)) represents the predicted salinity with a unit of ppt"

---

## Author Comment (AC2)

**Point-to-point actions in response to Anonymous Referee #2**

Manuscript #: essd-2025-671

We sincerely thank the Referee #2 for their thoughtful and constructive comments. We have revised the manuscript accordingly. The detailed point-by-point responses are provided below (blue), and the corresponding revisions in the manuscript are marked in red.

**Response to the Reviewer 2:**

The study fully leverages the physical mechanisms underlying passive remote sensing of water salinity and the high spatial resolution advantages of optical remote sensing, enabling salinity estimation for inland lakes across China. The lake salinity dataset provided by the research is highly valuable for monitoring biogeochemical processes within lake ecosystems. The manuscript is clearly structured, but a few minor issues still require revision. It could be considered for publication after these revisions are made. These small issues are as follows:

**Reply:** We are very thankful to the reviewer 2, and we appreciate their suggestions and valuable and positive comments for improving the manuscript! We have addressed to all comments to improve the quality of this manuscript.

1) Line 90: The citation order of the figures jumps from Fig. 1 directly to Fig. 10. You may need to move Fig. 10 to the position of Fig. 2, or remove the citation of Figure 10 at this point.

**Reply:** Thank you for your pointing this out. We have removed the inappropriate citation of Figure 10.

**(Page 3, lines 88-89)** "Notably, some years are missing data for Lake Hulun and Lake Juyan due to insufficient SAR imagery or multi-source data matching pairs."

2) Please move the word "Catchment" in Table 1 onto a single line to ensure the integrity of the term.

**Reply:** Thank you for your professional suggestions. We have revised it.

**(Page 4, lines 91-92)** "

Table 1

| Lake name | Coordinates | Catchment area (km²) | Hydrological connectivity | Date | Samples number | Dominant ions |
|---|---|---|---|---|---|---|

"

3) The section number 2.5 seems it should be changed to 2.6.

**Reply:** Thank you for your pointing this out! We have corrected the error and rechecked all serial numbers.

**(Page 8, lines 174)** "2.6 Construction of salinity model"

4) In the caption of Figure 3, please remove the comma ',' after "RMSE = 0.82 ppt".

**Reply:** Thank you for your pointing this out! We have revised it.

**(Page 14, lines 298-300)** "Figure 1: (a)-(i) Scatter plots of 30% test data (N = 84) for XGB ($R^2$ = 0.98 and RMSE = 0.58 ppt), RFR ($R^2$ = 0.97 and RMSE = 0.87 ppt), DNN ($R^2$ = 0.97 and RMSE = 0.82 ppt), CNN ($R^2$ = 0.98 and RMSE = 0.79 ppt), and stacking model ($R^2$ = 0.98 and RMSE = 0.60 ppt)"

5) Line 443: Should the square brackets be changed to parentheses?

**Reply:** Thank you for your professional comments. We have made the modification, replacing square brackets with parentheses.

**(Page 27, lines 445-446)** "The K&S model was designed for seawater, and its application to complex ionic lakes (e.g., Lake Chagannaoer) introduces about 5% uncertainty (Figure6 (f7))."

---

## Author Response (AR2)

**Point-to-point actions in response to Reviewers**

We sincerely thank the Referee for their thoughtful and constructive comments. We have revised the manuscript accordingly. The detailed point-by-point responses are provided below (blue), and the corresponding revisions in the manuscript are marked in red.

**Response to the Reviewer #1:**

The manuscript I reviewed is already a revised version. I like it very much. Not only is the research original, but the data products are unique and significant for various stakeholders. To my knowledge no one has published similar datasets in the literature. It's a pleasant read and I also learned from this reading. I have only a few editorial comments, which can be made during galley proof corrections.

**Reply:** We are very grateful that you took the time to review our manuscript, and sincerely thank your interest in our articles! Your valuable comments provided great assistance for the improvement of our articles. We addressed each comment as follows.

1) Fig. 1. Need to explain (a) in the caption. Also explain the meaning of "frequency of surface water occurrence" and the cross symbols in each panel.

**Reply:** Thank you for your professional suggestions. Firstly, we have added an explanation for Figure 1(a) in the figure caption. Secondly, we have explained that the frequency of surface water occurrence refers to the normalized count of pixels that were identified as water in Landsat imagery from 1984 to 2021. Finally, we have described the cross symbols in each panel, specifying that they represent field salinity sampling points.

**(Page 5, lines 93-97)** "Figure 1: (a) Spatial distribution of sampled lakes covered by the dataset, inlet rivers, and sub-basins. (b)-(l) Eleven study lakes in the IMXL from east to west and its surface water occurrence frequency (Pekel et al., 2016), including Lake Hulun, Lake Dalinor, Lake Chagannaoer, Lake Daihai, Lake Nanhaizi, Lake Hongjiannao, Lake Ulansuhai, Lake Juyan, Lake Ulungur, Lake Bosten, and Lake Sayram. Cross symbols in each lake denote the field salinity sampling points. The

frequency of surface water occurrence refers to the normalized count of pixels that were identified as water in Landsat imagery from 1984 to 2021."

2) Fig. 2 caption: Please add one sentence similar to this: "The details of individual steps and the meaning of the symbols are described below."

**Reply:** Thank you for your pointing this out. We have added it.

**(Page 9, lines 181-183)** "Figure 2: A brand-new lake water salinity estimation framework by a stacking salinity model, consisting of three steps: step one, data processing and feature construction; step two, model construction and ensemble; and step three, salinity estimation. Details of individual steps and the meaning of the symbols are described below."

3) Figs. 3 and 13. Change "Measure Salinity (ppt)" to "Measured Salinity (ppt)" in all x-axis labels.

**Reply:** Thank you for your professional suggestions. We have revised it.

**(Page 14, lines 298-301; Page 27, lines 498-499)**

[Figure]

**Figure 3**

**Figure 13**

4) Fig. 6. I like this figure as it makes visual inspection straightforward. The lower salinity around the land-lake boundaries may be due to runoff, but what caused the high salinity points (reddish colors) in c7?

**Reply:** We greatly appreciate your recognition, and thank you for your thoughtful comments. Through observation and analysis of MSI natural color composite images and salinity maps derived from various models, we found that the high salinity in the land-lake boundaries of Lake Bosten was caused by DNN and CNN overestimates of salinity influenced by mixed pixels. This high value was not fully corrected by the integrated model. Additionally, we supplemented the explanation for the high salinity pixels appearing in Figure 6(c7) in Section 3.2.

[Figure]

MSI natural color composite images

[Figure]

Salinity maps derived from different models

**(Page 16, lines 314-316)** "However, salinity values in Lake Bosten's nearshore pixels were not fully corrected because the DNN and CNN models overestimate salinity influenced by mixed pixels (Figure 6(c4,c5,c7))."

5) Fig. 11. Put some space between column 3 and column 4, and between column 6 and column 7.

**Reply:** Thank you for your pointing this out. We have revised it.

**(Page 25, lines 404-406)**

[Figure]

**Figure 11**

6) Eq. (8) – what's the unit of g? Need to list.

**Reply:** Thank you for your thoughtful comments. We have clearly specified that the units of integrated model estimates are aligned with the input data with units of ppt.

**(Page 12, lines 257-258)** "where g(E(Y)) represents the predicted salinity with a unit of ppt"

**Response to the Reviewer #2:**

The study fully leverages the physical mechanisms underlying passive remote sensing of water salinity and the high spatial resolution advantages of optical remote sensing,

enabling salinity estimation for inland lakes across China. The lake salinity dataset provided by the research is highly valuable for monitoring biogeochemical processes within lake ecosystems. The manuscript is clearly structured, but a few minor issues still require revision. It could be considered for publication after these revisions are made. These small issues are as follows:

**Reply:** We are very thankful to the reviewer 2, and we appreciate their suggestions and valuable and positive comments for improving the manuscript! We have addressed to all comments to improve the quality of this manuscript.

1) Line 90: The citation order of the figures jumps from Fig. 1 directly to Fig. 10. You may need to move Fig. 10 to the position of Fig. 2, or remove the citation of Figure 10 at this point.

**Reply:** Thank you for your pointing this out. We have removed the inappropriate citation of Figure 10.

**(Page 3, lines 88-89)** "Notably, some years are missing data for Lake Hulun and Lake Juyan due to insufficient SAR imagery or multi-source data matching pairs."

2) Please move the word "Catchment" in Table 1 onto a single line to ensure the integrity of the term.

**Reply:** Thank you for your professional suggestions. We have revised it.

**(Page 4, lines 91-92)** "

Table 1

| Lake name | Coordinates | Catchment area (km$^2$) | Hydrological connectivity | Date | Samples number | Dominant ions |
| --- | --- | --- | --- | --- | --- | --- |

"

3) The section number 2.5 seems it should be changed to 2.6.

**Reply:** Thank you for your pointing this out! We have corrected the error and rechecked all serial numbers.

**(Page 8, lines 174)** "2.6 Construction of salinity model"

4) In the caption of Figure 3, please remove the comma ',' after "RMSE = 0.82 ppt".

**Reply:** Thank you for your pointing this out! We have revised it.

**(Page 14, lines 298-300)** "Figure 1: (a)-(i) Scatter plots of 30% test data (N = 84) for XGB ($R^2$ = 0.98 and RMSE = 0.58 ppt), RFR ($R^2$ = 0.97 and RMSE = 0.87 ppt), DNN ($R^2$ = 0.97 and RMSE = 0.82 ppt), CNN ($R^2$ = 0.98 and RMSE = 0.79 ppt), and stacking model ($R^2$ = 0.98 and RMSE = 0.60 ppt)"

5) Line 443: Should the square brackets be changed to parentheses?

**Reply:** Thank you for your professional comments. We have made the modification, replacing square brackets with parentheses.

**(Page 27, lines 445-446)** "The K&S model was designed for seawater, and its application to complex ionic lakes (e.g., Lake Chagannaoer) introduces about 5% uncertainty (Figure6 (f7))."

**Response to the Reviewer #3:**

This study provides new insights into efficient monitoring of lake salinity for smaller lakes, due to improved spatial resolution achieved through the use of microwave and optical imagery. The presented methodology and accompanying data are of great value in environmental monitoring. The methods are described very clearly. The interpretation of the results and the visual representation are of high quality. Overall, I

consider this paper acceptable as it is, as I have read the already revised version and the comments of other referees.

**Reply:** We sincerely thank the reviewers for the positive feedback! We greatly appreciate the recognition of the methodology, interpretation of results, and visualizations presented in our work, as well as the significant value of the accompanying data for environmental monitoring.

**List:**

| Revised point | Brief description |
|---|---|
| Abstract | Modify asset DOI, "(https://doi.org/10.5281/zenodo.18371515)" |
| Dataset coverage | Added explanation |
| Table 1 | Modify the format |
| Figure 1 | Add additional content to the figure caption |
| 2.6 Construction of salinity model | Revision Number |
| 2.6 Construction of salinity model | Revision of grammar |
| Figure 2 | Modify the figure caption |
| 2.6.1 Mechanistic salinity model | Modify equation (2) and (4) |
| 2.6.1 Mechanistic salinity model | Modify equation (7) and parameter description |
| 2.6.3 Stacking salinity model | Additional explanation regarding the parameters of formula 8 |
| Figure 3 | Redraw and modify the figure caption |
| 3.2 Single-scene analysis of different models | Additional explanation content |
| 3.3 Comparison with previous single-satellite algorithms | Modify citation format |
| Figure 11 | Redraw and modify the figure caption |
| 4 Data availability | Modify asset references and descriptions |
| Section 5.1 | Enhance the language |
| Figure 13 | Redraw |
| References | Update References |